# Efficient Purification of R-phycoerythrin from Marine Algae (*Porphyra yezoensis*) Based on a Deep Eutectic Solvents Aqueous Two-Phase System

**DOI:** 10.3390/md18120618

**Published:** 2020-12-04

**Authors:** Yifeng Xu, Quanfu Wang, Yanhua Hou

**Affiliations:** 1School of Environment, Harbin Institute of Technology, Harbin 150090, China; 16B927074@stu.hit.edu.cn; 2School of Marine Science and Technology, Harbin Institute of Technology, Weihai 264209, China

**Keywords:** aqueous two-phase system, deep eutectic solvent, extraction, marine algae, R-phycoerythrin

## Abstract

R-phycoerythrin (R-PE), a marine bioactive protein, is abundant in *Porphyra yezoensis* with high protein content. In this study, R-PE was purified using a deep eutectic solvents aqueous two-phase system (DES-ATPS), combined with ammonium sulphate precipitation, and characterized by certain techniques. Firstly, choline chloride-urea (ChCl-U) was selected as the suitable DES to form ATPS for R-PE extraction. Then, single-factor experiments were conducted: the purity (A_565_/A_280_) of R-PE was 3.825, and the yield was 69.99% (*w/w*) under optimal conditions (adding 0.040 mg R-PE to ChCl-U (0.35 g)/K_2_HPO_4_ (0.8 g/mL, 0.5 mL) and extracting for 20 min). The sodium dodecyl sulfate-polyacrylamide gel electrophoresis (SDS-PAGE) results revealed that the purified R-PE contained three main bands. One band was presented after purification in native-PAGE. The UV-vis spectra showed characteristic absorption peaks at 495, 540, and 565 nm. R-PE displayed an emission wavelength at 570 nm when excited at 495 nm. All spectra results illustrated that the structure of R-PE remained unchanged throughout the process, proving the effectiveness of this method. Transmission electron microscope (TEM) showed that aggregation and surrounding phenomena were the driving forces for R-PE extraction. This study could provide a green and simple purification method of R-PE in drug development.

## 1. Introduction

R-phycoerythrin (R-PE) is a marine bioactive protein, mainly found in *Rhodophyta*, such as *Grateloupia turuturu* [1], *Heterosiphonia japonica* [2], and *Porphyra yezoensis*. Among them, *Porphyra yezoensis* is one of the major seaweeds abundantly cultivated in China [3]. It is easy to obtain, and the price is low. Besides, *P. yezoensis* has a higher protein content than other red algae (47% of dry weight) [4,5]. Because of this, it has attracted increasing attention as a healthy food and as a medicine to slow down the ageing process [3,6]. R-phycoerythrin (R-PE) is one of the major phycobiliproteins in *P. yezoensis*. Its purity is generally calculated by the absorbance ratio (A_565_/A_280_), and the market price is around 200 USD/mg (Sigma-Aldrich, product number: 52412-1MG-F) [7,8]. Therefore, R-PE is regarded as a bioactive molecule with high value. It has been reported that R-PE has numerous biological activities, including antioxidant [9], anti-inflammatory [10], anti-aging [11], and immunomodulatory activity [12,13]. R-PE has excellent optical properties as a natural fluorescent protein [1,14]. Based on these properties, R-PE has been widely applied in the biotechnological and medical fields. For example, it is used to treat cancer in photodynamic therapy as a model photosensitizer [15]. It is also used in immunology and cell biology and flow cytometry as a fluorescent label [16,17,18]. In addition, it is utilized in the food and cosmetics fields as a natural pigment [19]. In our previous study, R-PE was applied as a natural fluorescent probe to monitor mercury ions in water environments [20].

In view of the broad application prospects of R-PE, many conventional methods have been used for the extraction and purification of R-PE, such as ammonium sulfate precipitation combining with diethylaminoethyl (DEAE)–Sepharose Fast Flow column chromatography [1], hydroxyapatite column chromatography [14], and Q-Sepharose column chromatography [21]. These methods have their advantages and contribute in their own ways to R-PE purification yield. However, these purification procedures are usually needed high cost. In recent years, the aqueous two-phase system (ATPS), as a liquid–liquid extraction system, has offered an active alternative for extracting phycobiliprotein with high purity or improved extraction efficiency [22,23]. Usually, ATPS is composed of two polymers or a polymer and a salt in specific concentrations [24]. It can promise extraction of the desired substance in a relatively mild environment to ensure molecular stability throughout the extraction process.

With the development of green chemistry, green solvents have attracted notable attention. Deep eutectic solvents (DESs), as green solvents, are composed of a proper molar ratio of a hydrogen-bond acceptor (quaternary ammonium salts) and hydrogen-bond donors (acid amides, carboxylic acids, and polyhydric alcohols) [25,26]. As analogues of ionic liquids (ILs), DESs share many physicochemical properties with ILs, such as non-flammability, and excellent thermal and chemical stability [25]. DESs also have unique advantages, such as ease of preparation, and lower price and toxicity [27,28]. Therefore, DESs have been applied in the fields of electrochemistry [29], catalysts [30], and extraction of bioactive compounds, such as polyphenols [31], catechins [32], and proteins [33].

Based on the high commercial value and great application potential in food and medical field of R-PE, a low-cost purification method needed to be developed. To the best of our knowledge, there was no report on purifying R-PE using green solvent DES combined with ATPS. Therefore, the main focus of this study was to develop an environmentally friendly and low-cost method combining a green DES with ATPS, coupled with ammonium sulfate precipitation and applied to extract R-PE from the natural marine algae *P. yezoensis*, which had high protein content. Additionally, its properties and purification mechanisms were identified. The drug-grade R-PE (purity was 3.0) was obtained, and the method was cheaper.

## 2. Results and Discussion

### 2.1. Characterization of DESs

The structures of the obtained DESs are shown in Appendix A. The absorption bands of all DESs exhibited a remarkably broad and strong peak between 3300–3400 cm^−1^, contributing to the OH stretching bands at approximately 3300 cm^−1^, which represented the presence of abundant hydrogen bonds in the six kinds of DESs.

### 2.2. Establishment and Phase Behavior of DES-Based ATPS

The data of the DES-based phase diagram determined at 298.15 K and atmospheric pressure are shown in Figure 1. K_2_HPO_4_ was selected as the phase separation salt because of its strong phase-forming ability and high solubility in water [34]. According to the former literature report, if the phase diagram of DES-K_2_HPO_4_ is close to the origin of the coordinate axis, the phase-forming ability of the DES is stronger [25]. Therefore, the phase-forming abilities of the DESs were in following descending sequence: choline chloride-urea (ChCl-U) > choline chloride-D-glucose (D-ChCl-Gl) > choline chloride-glycerol (ChCl-G) > choline chloride-D-fructose (D-ChCl-F) > choline chloride-D-sorbitol (D-ChCl-S) > choline chloride-ethylene glycol (ChCl-EG). It was presumed that the difference in abilities of these DESs to form phase was related to their affinity for water molecules [35,36]. The illustration corresponded to the sequence of the phase diagram from left to right: ChCl-U, D-ChCl-Gl, ChCl-G, D-ChCl-F, D-ChCl-S, and ChCl-EG.

### 2.3. Selection of DES-ATPS for the R-PE Extraction

The DES-ATPSs above were investigated for R-PE extraction (as shown in Figure 2). Taking purity and extraction efficiency as the indicators, ChCl-U/K_2_HPO_4_ was the best system for R-PE purification among the six ATPSs. Therefore, ChCl-U/K_2_HPO_4_ was chosen as the extraction system for the subsequent study.

### 2.4. Single-Factor Experiments

To ascertain the impact of the amount of DES on the extraction efficiency and purity of R-PE, different amounts of ChCl-U (0.25–0.55 g) were mixed into six copies of the K_2_HPO_4_ solution (0.8 g/mL, 0.5 mL) to compose ATPS. The same volume of dialyzed R-PE was added to the ATPSs and mixed the systems again. After phase separation, the extraction efficiency and purity could be obtained by measuring the absorbance at the corresponding wavelengths and calculating by the relevant equations. It was discovered that 0.35 g was the most suitable amount of DES for the optimal extraction conditions (Figure 3a). This trend was because an increased amount of DES resulted in high viscosity of the top phase, preventing R-PE from transferring into the top phase [35]. This result was similar to that for ChCl-U/K_2_HPO_4_ (0.7 g/mL, 2.0 mL, 1.4 g) [37] and betaine-urea (Be-U)/K_2_HPO_4_ (0.75 g/mL, 2.0 mL, 1.4 g) [38] and higher than that of choline chloride-glycerol (ChCl-G)/K_2_HPO_4_ (0.9 g/mL, 2.0 mL, 1.3 g) [35], found in previous studies.

It was clear that the extraction efficiency and purity increased with the increasing amount of R-PE lower than 0.040 mg and decreased when the amount of R-PE was higher than 0.040 mg (Figure 3b). Therefore, the optimum amount of R-PE was 0.040 mg. The possible reason for this phenomenon was that an increase in the amount of R-PE led to high extraction efficiency and purity, and with a further increase in the amount of R-PE, the extraction system approached saturation, leading to extraction efficiency and purity reduction [38].

As shown in Figure 3c, the maximal extraction efficiency and purity were acquired when the extraction times were 30 and 20 min, respectively. Considering extraction efficiency, purity, and energy saving, the optimal extraction time was 20 min. With the increase in time, the R-PE in the top phase escaped from the package of the DES to the bottom phase, leading to a decline of the extraction efficiency and purity [35]. Under optimal conditions, the extraction efficiency of R-PE could reach 92.60%, and the purity of R-PE reached 3.825 (the ratio of A_565_/A_280_), up to drug-grade (3.0).

With the aim of validating feasibility of the method, precision, repeatability, and stability experiments were conducted (Appendix A). The relative standard deviations (RSDs) of three experiments were 0.92, 0.85, and 0.73% (*n* = 5) (<10%), respectively, which indicated that the method had acceptable feasibility.

### 2.5. Electrophoresis Assay of Purified R-PE

The SDS-PAGE result of the fractions collected during the purification steps is shown in Figure 4. It revealed that impurities were almost completely removed after purification in the ATPS, proving the effectiveness of this purification method. It can also be seen from the SDS-PAGE, as shown by zinc acetate staining, that the purified R-PE contained three fluorescent bands (Figure 4a, lane 7 and 8), representing the formation of fluorescent chelates between the phycobilirubin of R-PE and zinc ions. Moreover, it was found that three subunits of R-PE (α, β and γ subunits) appeared in the same position in the SDS-PAGE, as shown by Coomassie blue R-250 staining (Figure 4b, lane 7 and 8), whose molecular masses were about 18.0, 21.0, and 30.0 kDa. The previous report confirmed the presence of three similar characteristic subunits of about 17, 19, and 33 kDa in R-PE from *P. yezoensis* [39].

The results of native-PAGE, as shown by the zinc acetate staining (Figure 5a) and Coomassie blue R-250 staining (Figure 5b), showed that the components of fluorescent protein gradually presented one band during the whole purification process. As shown in Figure 5b, only one band was presented as well (lane 7 and 8), indicating that the purified R-PE was homogenous.

### 2.6. R-PE Purity and Yield Analysis

The total protein content, R-PE content, yield, and purity after each purification step were calculated and are shown in Table 1, in order to study the change in R-PE during the whole purification process. It can be seen that the purity increased and R-PE content, protein content, and yield decreased due to loss in the purification process. The purity of R-PE could be improved from 0.713 to 3.825, and the extraction yield for the DES phase in relation to the first frozen-thawing extract was 69.99%.

### 2.7. Characterization of R-PE Properties and Mechanism of Purification Method Based on DES-ATPS

#### 2.7.1. Characterization of R-PE Optical Properties before and after Purification

UV-vis spectra and fluorescence spectra were measured to characterize the spectral properties of R-PE. As shown in Figure 6a, R-PE had two absorbance peaks at 495 and 565 nm, and a shoulder at 540 nm. R-PE exhibited an excitation wavelength at 495 nm and emission wavelength at 570 nm (Figure 6b). These optical properties of R-PE were in accordance with those of R-PE purified by conventional purification methods such as hydroxyapatite chromatography [20] and DEAE–Sepharose Fast Flow chromatography [1], which could prove the effectiveness of DES-ATPS. Further, the spectra followed a similar trend before and after purification. These results suggested that there were neither interaction effects nor chemical bonds formed between the R-PE and DESs, which further demonstrated that the spatial structure of R-PE was not destroyed [40].

The use of FT-IR spectra is to investigate functional groups or chemical bonds in a molecule of an interaction system. Figure 7 (black and blue lines) indicates that the absorption bands of DES (1472 cm^−1^) and dialyzed R-PE (1074 cm^−1^) were recognizable in the spectra before extraction. After extraction, these two bands were still identifiable in the spectra of R-PE-DES complexity (red line). This suggested that the functional groups of R-PE were maintained well, indicating that the structure of the R-PE was not changed. In the previous studies, DES-based ATPS, such as ChCl-glycerol/K_2_HPO_4_ [35] and betaine-urea/K_2_HPO_4_ [41], also maintained the structure and activity of the protein during the extraction process, which made such systems competitive candidates in the field of extracting proteins.

#### 2.7.2. Characterization of R-PE by Circular Dichroism Spectra before and after Purification

Circular dichroism (CD) spectra were measured to characterize the secondary structure of dialyzed R-PE and R-PE in the DES-rich top phase. As shown in Figure 8, the CD spectra of dialyzed R-PE showed negative ellipticity between 202 and 240 nm (minima 208 and 222 nm) and positive ellipticity at 192 nm, corresponding to the CD spectral characteristics of α-helical proteins [42]. It was deduced that R-PE mainly contains α-helix, and the results were in accordance with previous reports [43]. The CD spectra curve of dialyzed R-PE was similar to that of R-PE in the DES-rich top phase, which indicated that the secondary structure of R-PE was maintained well during the purification process.

#### 2.7.3. Mechanism of the Purification Method Based on DES-ATPS

In order to further study the driving forces of the extraction process, TEM was carried out to examine the morphology of the DES and dialyzed R-PE (before extraction and in the DES-rich top phase after extraction). The size of the dialyzed R-PE was about 800 nm. The conformation of the DES-rich top phase without R-PE was in a state of less aggregation (Figure 9b). After adding the dialyzed R-PE extract, the DES encircled the R-PE particles and the complex took shape (Figure 9c). DES could form DES micelles to aggregate R-PE to the DES-rich top phase [38]. Therefore, it was deduced that the “aggregation and surrounding phenomenon” [33] were the main driving force in the extraction process by the DES-based ATPS. The results were similar to those reported in previous literature [38].

### 2.8. Comparison with Other Purification Methods

For comparative purpose, Table 2 summarizes several reported methods for R-PE extraction. In this study, the purity of R-PE was higher than that achieved by DEAE–Sepharose Fast Flow chromatography [1] and ATPS (PEG 1450/K_3_PO_4_) [44], lower than that of chromatography combined with other pretreatment methods [2,21,45], and close to that of hydroxyapatite column chromatography [14]. The extraction yield of the present study was lower than for ATPS (PEG 1450/K_3_PO_4_) [44] and higher than for other methods mentioned in Table 2. In this research, the developed DES-ATPS had the advantages of being green and simple, which makes it a promising alternative for extracting water-soluble components including value-added R-PE.

## 3. Materials and Methods

### 3.1. Materials

Choline chloride (C_5_H_14_ClNO) (dried in a DZF-6051 vacuum drying oven (Shanghai, China) for about 2 h at 110 °C before use), urea (CO(NH_2_)_2_), D-sorbitol (C_6_H_14_O_6_), D-(+)-glucose (C_6_H_12_O_6_), ethylene glycol ((CH_2_OH)_2_), D-(-)-fructose (C_6_H_12_O_6_), glycerol (C_3_H_8_O_3_), and K_2_HPO_4_ were purchased from Sinopharm Chemical Reagent Co., Ltd. All of these reagents were of analytical grade. Dried seaweed (*P. yezoensis*) was purchased from Yantai, Shandong Province, China. Ultrafiltration centrifuge tube (MWCO 10 kDa) was purchased from Merck KGaA (Darmstadt, Germany). Protein Molecular Weight Marker 3450Q (Takara Biomedical Technology (Beijing) Co., Ltd., Beijing, China) was used as the molecular weight marker.

### 3.2. R-PE Pre-Treatment from P. yezoensis

A total of 5.0 g of *P. yezoensis* was weighed and washed with phosphate buffer (PB, 10 mM, pH 6.8) and repeatedly frozen and thawed four times (−25 °C and 4 °C) [39]. The obtained slurry was filtrated by gauze, then centrifuged (7500× *g*, 20 min, 4 °C); this was the crude extract. The crude extract was precipitated with (NH_4_)_2_SO_4_ at saturation of 10% (5.5 g (NH_4_)_2_SO_4_ added per 100 mL of the crude extract). After it was placed in static conditions for 8 h at 4 °C, it was centrifuged (7500× *g*, 20 min, 4 °C) to obtain the supernatant. Then, the supernatant was precipitated with (NH_4_)_2_SO_4_ at a saturation of 50% (25.1 g (NH_4_)_2_SO_4_ added per 100 mL of the supernatant) and left for 8 h at 4 °C. With the same centrifuge process mentioned above, the precipitate was obtained. The final precipitate was dissolved and dialyzed with an MWCO 8000 Da dialysis cassette against 500 mL of PB solution (10 mM, pH 6.8), and the fresh buffer was changed every half hour. After about six hours, 0.5 mL dialysate was taken into a 1.5 mL centrifuge tube, and 0.5 mL 0.1 M BaCl_2_ was added. No white precipitation production indicated that there was no (NH_4_)_2_SO_4_ in the dialysate, implying the end of the dialysis process. The obtained solution was called dialyzed R-PE.

### 3.3. Deep Eutectic Solvents Preparation

Six kinds of deep eutectic solvents (DES) were synthesized by stirring the eutectic mixtures at 80 °C for about 2 h, until a homogenous liquid was formed. The ratios of quaternary ammonium salts (choline chloride) to hydrogen bond donors (Ethylene glycol, D-sorbitol, glycerol, D-(+)-glucose, D-fructose, and urea) were 1:2, 1:1, 1:2, 2:1, 1.9:1, and 1:2, respectively [35,46]. The structures of the obtained DESs were confirmed by an FT-IR spectrometer (TENSOR, Bruker Optics, Karlsruhe, Germany).

### 3.4. Establishment and Phase Behavior Study of the DES-Based ATPS

For the phase diagrams determination, the cloud point titration method was used, based on atmospheric pressure and room temperature [47]. A total of 2.0 g of the DES was weighed into a 10 mL centrifuge tube, then 1.0 g/mL of K_2_HPO_4_ solution was added drop by drop to the centrifuge tube and shaken until the mixture became cloudy, indicating DES/K_2_HPO_4_ ATPS formation. The mass of K_2_HPO_4_ solution added was recorded. Then, a certain volume of ultrapure water was added to the system dropwise until phase one occurred. The mass of water was also recorded after its addition (the mass difference of the system before and after adding the water). Finally, the process above was repeated to construct the phase diagrams of liquid–liquid equilibrium. The X axis and Y axis represented the mass fraction of the K_2_HPO_4_ solution and DES, respectively.

### 3.5. Selection of DES-ATPS for the R-PE Extraction

In order to select the suitable DES-ATPS for R-PE extraction, 0.35 g of six kinds of DESs were weighed into six 1.5 mL centrifuge tubes and of K_2_HPO_4_ solution (0.8 g/mL, 0.5 mL) were added into the centrifuge tubes to compose ATPS. Then, six copies of the same mass of dialyzed R-PE (0.040 mg) were added into ATPSs. The extraction efficiency and purity were calculated using the following equations from the absorbance measured at the corresponding wavelengths.

The efficiency of R-PE in the two-phase system was calculated by the following equations:(1)$$E=\left(C_{t}V_{t}\right)/\left(C_{t}V_{t}+C_{b}V_{b}\right)$$
where C_t_ and C_b_ are the concentrations of the R-PE in the DES-rich top phase and salt-rich bottom phase, respectively. V_t_ and V_b_ represent the volume of the top phase and the bottom phase, respectively.

Purity was calculated by the following formula [45]:(2)$$\mathrm{Purity}=A_{565}/A_{280}$$
where A_565_ and A_280_ are the absorbance of R-PE at 280 and 565 nm, respectively.

### 3.6. Single-Factor Experiments

Based on the selected ATPS above, single-factor experiments were performed by varying one factor varied at different levels, while the other factors were fixed, in order to explore the optimum conditions for the extraction efficiency (E) and purity of R-PE. First of all, different amount of DES (0.25, 0.30, 0.35, 0.40, 0.45, 0.50, and 0.55 g) were added into six copies of the K_2_HPO_4_ solution (0.8 g/mL, 0.5 mL) to compose ATPS. A certain mass of dialyzed R-PE was added to the system. After phase separation, the top phase was filtered through ultrafiltration centrifuge tube (MWCO 10 kDa) (4500× *g*, 10 min, 4 °C) to obtain purified R-PE, which was needed in the following analysis. The extraction efficiency and purity were calculated using the relevant equations. Then, the influence of different amounts of dialyzed R-PE (0.035, 0.040, 0.045, 0.050, 0.055, and 0.060 mg) and extraction times (0, 10, 20, 30, 40, 50, and 60 min) on extraction efficiency and purity were studied in the same way, respectively.

Under the selected optimal conditions above, precision, repeatability, and stability experiments were accomplished to validate the feasibility of the method. Apparatus precision was determined through five parallel measurements of the top phase under the same conditions. The repeatability was validated by taking five copies of one sample under the same conditions. The stability was determined over five consecutive days under the same conditions.

### 3.7. Electrophoresis Assay of Purified R-PE

SDS-PAGE was carried out using 15% separating and 5% stacking gel based on a previous study with some modification (70 V voltage for the initial 20 min and then 85 V until the end) [48]. Native-PAGE was performed with 10% separating and 5% stacking gel according to the previous literature with some modification (70 V for the initial 20 min and then 85 V until the end) [49]. After gel electrophoresis, the gel was soaked in 20 mM zinc acetate solution for 5 min at room temperature. After rinsing with double-distilled water four times (1–2 min each time), the gel was stained by Coomassie blue R-250 (0.1%, *w*/*v*). The gel electrophoresis results were analyzed by the ChemiDoc XRS+ system (Bio-Rad, Berkeley, CA, USA).

### 3.8. R-PE Purity and Yield Analysis

In order to study the purity and yield change in R-PE throughout the whole purification process, the total protein content, R-PE content, yield, and purity after each purification step were measured and calculated. The total protein content was determined by the Bradford method. The purity, R-PE concentration, and yield were calculated by Equations (2)–(4).

The concentration of R-PE was measured using the method from a previous report [50] with some modifications:(3)$$R-\mathrm{phycoerythrin}\;\left(\mathrm{mg}/\mathrm{mL}\right)=\left(0.123A_{565}-0.068A_{618}+0.015A_{650}\right)\times n$$
where A_565_, A_618_, and A_650_ are the absorbance of R-PE at 565, 618, and 650 nm, respectively. *n* represents the dilution fold.

The R-PE yield from total protein was calculated by the following equation:(4)$$R-\mathrm{PE}\;\mathrm{yield}=\frac{R-\mathrm{PE}\;\mathrm{content}\;\mathrm{after}\;\mathrm{purification}}{R-\mathrm{PE}\;\mathrm{content}\;\mathrm{in}\;\mathrm{first}\;\mathrm{frozen}-\mathrm{thawing}\;\mathrm{extract}}$$

### 3.9. R-PE Properties Characterization and Measurement of the Extraction Mechanism

The UV spectra of the purified samples were obtained using a UV-vis spectrophotometer (UV2450, Shimadzu, Kyoto, Japan). Fluorescence spectra of the samples were measured using a fluorescence spectrometer (F-2700, Hitachi, Tokyo, Japan). FTIR spectra were recorded at room temperature on an FTIR spectrometer (TENSOR, Bruker Optics, Karlsruhe, Germany). The secondary structure was measured using a Circular Dichroism (CD) Spectrometer (MOS-500, Bio-Logic, Paris, France). The microstructure of the samples was examined with a transmission electron microscope (JEOL-2100, JEOL, Tokyo, Japan). The experimental data analysis was performed by Origin 8.5.

### 3.10. Statistical Analysis

Statistical significance of the results was analyzed by Statistical Product and Service Solutions (SPSS) 19.0 software. Asterisks represented statistical significance (* *p* ≤ 0.05; ** *p* ≤ 0.01; ns, *p* > 0.05).

## 4. Conclusions

In view of the wide application of R-PE in the drug and food fields, in this work, the marine bioactive protein R-PE was purified from *P. yezoensis* using green DES combined with ATPS. Under the optimal conditions, a final yield of 69.99% and purity of 3.825 were achieved with this system. UV-vis, fluorescence, FT-IR spectra, and circular dichroism spectra illustrated that the spatial structure and secondary structure of R-PE were maintained throughout the whole process. The aggregation and embrace phenomena drove the extraction of R-PE in the separation process by TEM. This green and simple purification method based on DES-ATPS may offer new possibilities for obtaining more active drug molecules from marine algae.

## Figures and Tables

**Figure 1 marinedrugs-18-00618-f001:**
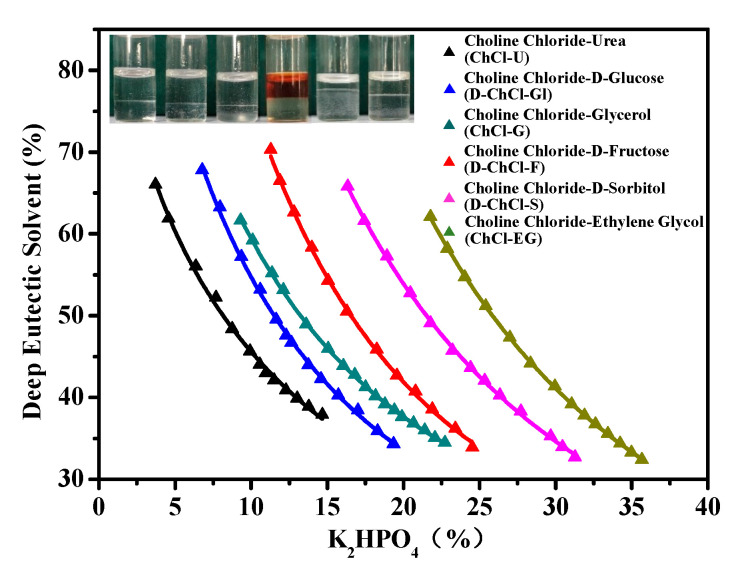
The phase diagrams of deep eutectic solvents (DESs) + K_2_HPO_4_ aqueous two-phase systems (ATPSs). ChCl-U represents choline chloride-urea, D-ChCl-Gl represents choline chloride-D-glucose, ChCl-G represents choline chloride-glycerol, D-ChCl-F represents choline chloride-D-fructose, D-ChCl-S represents choline chloride-D-sorbitol, ChCl-EG represents choline chloride-ethylene glycol. The illustration showed six ATPSs in phase diagram order.

**Figure 2 marinedrugs-18-00618-f002:**
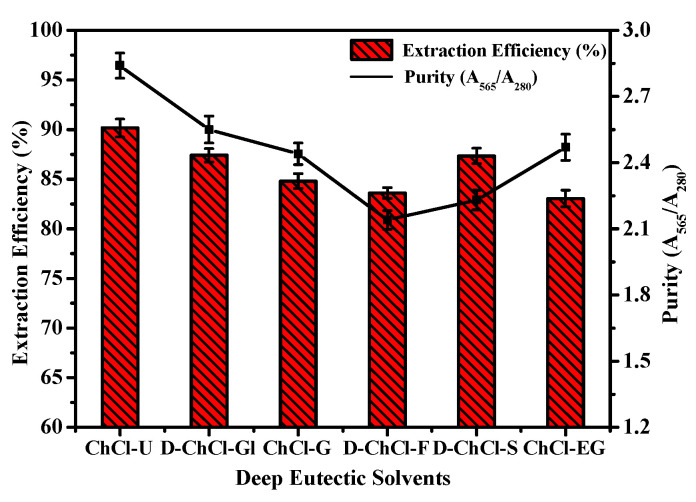
Effect of different deep eutectic solvents aqueous two-phase systems (DES-ATPSs) on the extraction efficiency (${E\;=\;(C}_{t}V_{t})/{(C}_{t}V_{t}{+C}_{b}V_{b})$) and purity (A_565_/A_280_). Bars donate standard deviation (repeating three times). ChCl-U represents choline chloride-urea, D-ChCl-Gl represents choline chloride-D-glucose, ChCl-G represents choline chloride-glycerol, D-ChCl-F represents choline chloride-D-fructose, D-ChCl-S represents choline chloride-D-sorbitol, ChCl-EG represents choline chloride-ethylene glycol.

**Figure 3 marinedrugs-18-00618-f003:**
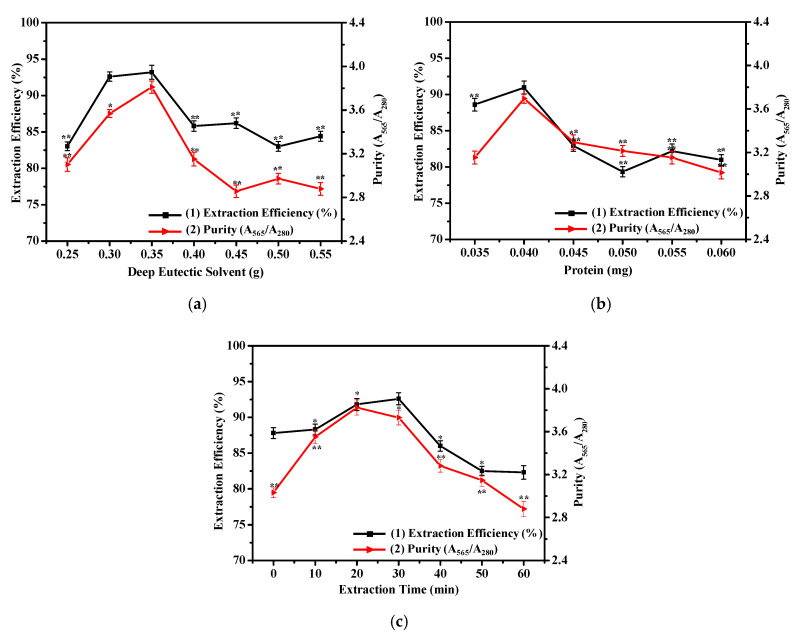
Effect of (**a**) amount of deep eutectic solvent (DES), (**b**) amount of R-PE, and (**c**) extraction time on the extraction efficiency (${E\;=\;(C}_{t}V_{t})/{(C}_{t}V_{t}{+C}_{b}V_{b})$) and purity (A_565_/A_280_) in the DES/K_2_HPO_4_ ATPS. Bars donate standard deviation (repeating three times). Asterisks represent statistical significance (* *p* ≤ 0.05; ** *p* ≤ 0.01; ns, *p* > 0.05).

**Figure 4 marinedrugs-18-00618-f004:**
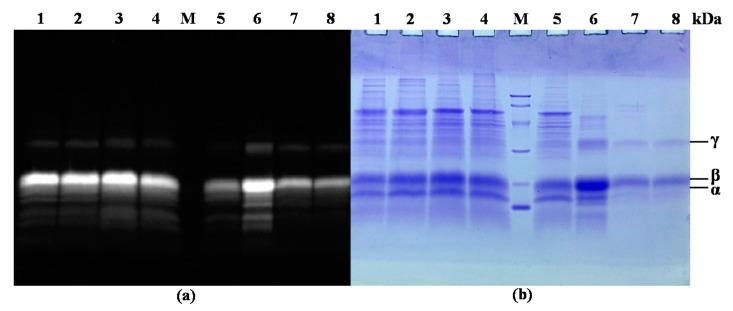
SDS-PAGE stained with zinc acetate (**a**) and Coomassie blue R-250 (**b**) of the fractions collected during the extraction and purification steps. Lanes 1–4, first to fourth frozen and thawing extract (crude extract); lane M, molecular mass markers (from top to bottom: 97.2, 66.4, 44.2, 29.0, 20.1, and 14.3 kDa); lane 5, first ammonium sulphate precipitation supernatant; lane 6, dialyzed R-PE; lane 7, purified R-PE by DES-ATPS (about 0.035 mg); lane 8, purified R-PE by DES-ATPS (about 0.030 mg).

**Figure 5 marinedrugs-18-00618-f005:**
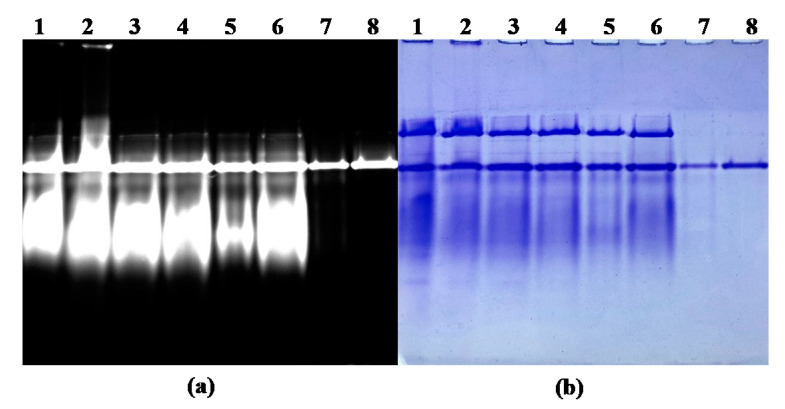
Native-PAGE stained with zinc acetate (**a**) and Coomassie blue R-250 (**b**) of the fractions during the extraction and purification steps. Lanes 1–4, first to fourth frozen and thawing extract (crude extract); lane 5, first ammonium sulphate precipitation supernatant; lane 6, dialyzed R-PE; lane 7, purified R-PE by DES-ATPS (about 0.030 mg); lane 8, purified R-PE by DES-ATPS (about 0.035 mg).

**Figure 6 marinedrugs-18-00618-f006:**
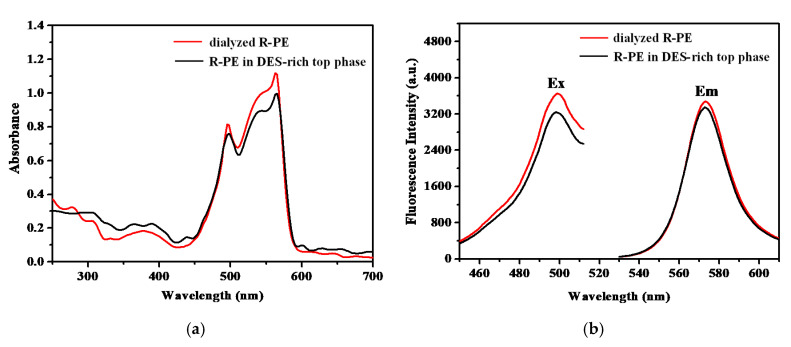
(**a**) UV-vis spectra of dialyzed R-PE and R-PE in deep eutectic solvent (DES)-rich top phase after extraction; (**b**) excitation wavelength (E_x_) and emission wavelength (E_m_) of dialyzed R-PE and R-PE in DES-rich top phase after extraction.

**Figure 7 marinedrugs-18-00618-f007:**
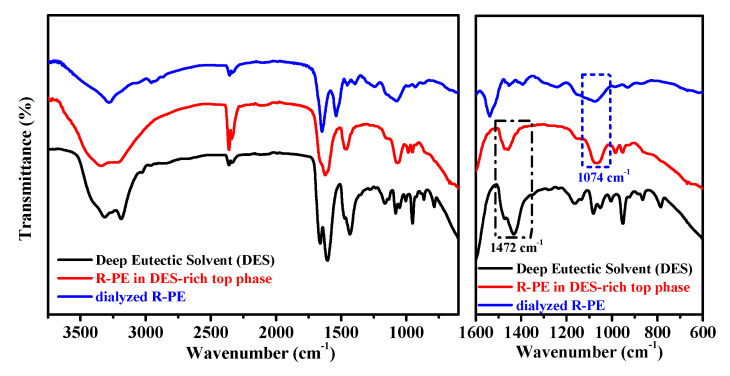
FT-IR spectra analysis of deep eutectic solvent (DES) (black line), R-PE in DES-rich top phase (red line), and dialyzed R-PE (blue line).

**Figure 8 marinedrugs-18-00618-f008:**
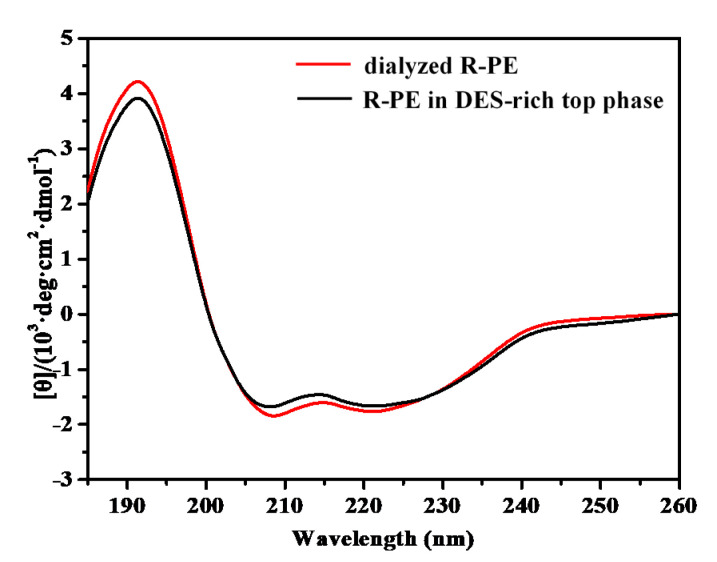
CD spectra of dialyzed R-PE and R-PE in deep eutectic solvent (DES)-rich top phase after extraction.

**Figure 9 marinedrugs-18-00618-f009:**
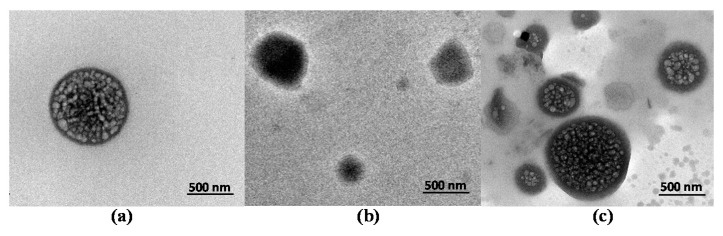
TEM images in the extraction process. (**a**) Dialyzed R-PE before extraction, (**b**) deep eutectic solvent (DES) before extraction, and (**c**) DES-rich top phase after extraction.

**Table 1 marinedrugs-18-00618-t001:** Purity and other index of R-PE from *P. yezoensis* after each stage of purification.

Purification Step	Total Protein Content (mg)	R-PE Content (mg)	Yield (%)	Purity (A_565_/A_280_)
First to fourth frozen-thawing extract (crude extract)	174.62	88.29	94.43	0.713
Two-step salting-out extraction (dialyzed R-PE)	96.94	70.68	75.59	2.293
R-PE after aqueous two-phase system (ATPS)	69.23	65.45	69.99	3.825

Total protein content was determined by the Bradford method; R-PE yield = (R-PE content after purification)/(R-PE content in first frozen-thawing extract).

**Table 2 marinedrugs-18-00618-t002:** Comparison of this work with various reported methods for R-PE extraction.

Method	Source	Yield (%)	Purity (A_565_/A_280_)	References
DEAE–Sepharose Fast Flow chromatography	*Grateloupia turuturu*	27.00	2.890	[1]
Gel filtration + DEAE–Sepharose Fast Flow chromatography	*Heterosiphonia japonica*	none	4.890	[2]
Ammonium sulfate precipitation +DEAE-Sepharose Fast Flow chromatography	*Polysiphonia urceolata*	67.33	5.600	[45]
Ammonium sulfate precipitation + Q-Sepharose column chromatography	*Portieria hornemannii*	64.80	5.210	[21]
Hydroxyapatite chromatography	*Polysiphonia urceolata*	22.00	3.900	[14]
Aqueous two-phase system (ATPS) (PEG 1450/K_3_PO_4_)	*Gelidium pusillum*	72.00	1.100	[44]
Deep eutectic solvent aqueous two-phase system (DES-ATPS) (choline chloride-urea/K_2_HPO_4_)	*P. yezoensis*	69.99	3.825	This work

R-PE yield = (R-PE content after purification)/(R-PE content in first frozen-thawing extract).

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
