# Peer review of "Efficient Purification of R-phycoerythrin from Marine Algae (Porphyra yezoensis) Based on a Deep Eutectic Solvents Aqueous Two-Phase System"

_marinedrugs, 2020, doi:10.3390/md18120618_

Round 1

Reviewer 1 Report

Although the authors attempted to revise their work, and it seems quite improved, many parts of the manuscript still need rewriting for sense and clarification. Figures and Tables should be understandable when they stand alone. Therefore, they should not contain any undefined abbreviation. Please correct and clarify throughout the manuscript. Some examples of my comments can be found below.

Title: Wrong. Please check how to use the right term for your system in the abstract.

Abstract:
Line 17: What is ChCl? Never seen; please use its full term.

Lines 17-18: ... as the suitable DES ... How did you know that it is suitable? You said 'was selected,' meaning that you tested other different couples of solvents? What were those tested? Please clarify.

Line 20: 69.99% in comparison to what, w/w, w/v, fresh weight, dry weight, which volume? Please clarify. And what were the optimal conditions?

Introduction:
Line 35: Remove '(P. yezoensis).'

Line 36: phylum is not in italic.

Line 41: No space in USD/mg.

Line 46: medical, not medicine.

Line 53: Provide full term of DEAE. Are sepharose chromatography here and sepharose column chromatography at line 54 the same? Check your writing.

Line 55: Remove the terms 'own.'

Line 69: A comma behind [31].

Line 72: What is an environmental method? Did you refer to an environmentally-friendly method?

Results and Discussion
Line 86: phase-forming.

Lines 98-105: The whole paragraph is unclear. The concentrations, compositions, and ratios of all solvents are unclear and hard to understand how to prepare them; please revise. All chemicals are written differently; please correct and use the same format.
ChCl-U/K2HPO4
ChCl-urea/k2HPO4
Betaine–Urea/K2HPO4
ChCl-glycerol/K2HPO4

Line 104: What did you mean by 'This'?

Lines 132-133: What do you mean by good RSDs, good precision, repeatability and stability? How good is it?

Author Response

Response to Reviewer 1 Comments

Point 1: Although the authors attempted to revise their work, and it seems quite improved, many parts of the manuscript still need rewriting for sense and clarification. Figures and Tables should be understandable when they stand alone. Therefore, they should not contain any undefined abbreviation. Please correct and clarify throughout the manuscript. Some examples of my comments can be found below.

Response 1: We appreciate very much for the Reviewer’s good comments and kind recommendation. We would try our best to give satisfactory answers in the revised version.

Besides, the explanations in figures and tables were modified more clearly in the revision (Please See, Page 3, Figure 1 and its legend; Page 3, Figure 2 and its legend; Page 4, Figure 3 and its legend; Page 6, Table 1 and legend of Figure 6; Page 7, Figure 7 and its legend, legend of Figure 8; Page 8, legend of Figure 9 and Table 2).

Point 2: Title: Wrong. Please check how to use the right term for your system in the abstract.

Response 2: We appreciate very much for the Reviewer’s good comments and suggestion. The title was checked and the expression for the system was unified to “deep eutectic solvent aqueous two-phase system” throughout the revised manuscript (Please See, Page 1, Line 15-16; Page 3, Line 103; Page 8, Table 2).

Abstract:

Point 3: Line 17: What is ChCl? Never seen; please use its full term.

Response 3: We appreciate very much for the Reviewer’s useful comment. It was revised in the revised version (Please See, Page 1, Line 17).

Point 4: Lines 17-18: ... as the suitable DES ... How did you know that it is suitable? You said 'was selected,' meaning that you tested other different couples of solvents? What were those tested? Please clarify.

Response 4: We appreciate very much for the Reviewer’s useful comment. Actually, choline chloride (ChCl)-urea was chosen as the suitable extraction solvent through the selection experiment. It was revised in the revised version (Please See, Page 3, Line 98-102, Figure 2 and Page 10, Line 287-298).

Point 5: Line 20: 69.99% in comparison to what, w/w, w/v, fresh weight, dry weight, which volume? Please clarify. And what were the optimal conditions?

Response 5: We appreciate very much for the Reviewer’s useful comments. It was revised in the revised version (Please See, Page 1, Line 20 and Line 20-21).

Introduction:

Point 6: Line 35: Remove '(P. yezoensis).'

Response 6: We appreciate very much for the Reviewer’s useful comment. It was modified in the revised version (Please See, Page 1, Line 35).

Point 7: Line 36: phylum is not in italic.

Response 7: We appreciate very much the very helpful comments. This was corrected in the revised version (Please See, Page 1, Line 35).

Point 8: Line 41: No space in USD/mg.

Response 8: We appreciate very much the very helpful comments. This was corrected in the revised version (Please See, Page 2, Line 41).

Point 9: Line 46: medical, not medicine.

Response 9: We appreciate very much the very careful comments. This was corrected in the revised version (Please See, Page 2, Line 46).

Point 10: Line 53: Provide full term of DEAE. Are sepharose chromatography here and sepharose column chromatography at line 54 the same? Check your writing.

Response 10: We appreciate very much for the Reviewer’s useful suggestion. It was amended in the revised version (Please See, Page 2, Line 53).

Point 11: Line 55: Remove the terms 'own.'

Response 11: We appreciate very much for the very helpful comments. It was deleted in the revised version (Please See, Page 2, Line 55).

Point 12: Line 69: A comma behind [31].

Response 12: We appreciate very much for the Reviewer’s carefulness. This was added in the revised version (Please See, Page 2, Line 70).

Point 13: Line 72: What is an environmental method? Did you refer to an environmentally-friendly method?

Response 13: We appreciate very much for the very helpful comments. It referred to “an environmentally-friendly method” and it was amended in the revised version (Please See, Page 2, Line 73).

Results and Discussion

Point 14: Line 86: phase-forming.

Response 14: We appreciate very much for the Reviewer’s useful suggestion. It was amended in the revised version (Please See, Page 2, Line 85).

Point 15: Lines 98-105: The whole paragraph is unclear. The concentrations, compositions, and ratios of all solvents are unclear and hard to understand how to prepare them; please revise. All chemicals are written differently; please correct and use the same format.

ChCl-U/K2HPO4

ChCl-urea/k2HPO4

Betaine–Urea/K2HPO4

ChCl-glycerol/K2HPO4

Response 15: We appreciate very much the very helpful comments. This part was redescribed clearer (Please See, Page 3, Line 111-115) and the format of chemicals was corrected in the revised version (Please See, Page 4, Line 118-120).

Point 16: Line 104: What did you mean by 'This'?

Response 16: We appreciate very much the very helpful comments. “This” meant the trend of the extraction efficiency and purity rose first and then decreased with the increase of DES (Please See, Page 4, Line 116-118).

Point 17: Lines 132-133: What do you mean by good RSDs, good precision, repeatability and stability? How good is it?

Response 17: We appreciate very much the very helpful comments. Relative standard deviation (RSD) is the percentage of the standard deviation to the average. It is generally used to evaluate the precision and repeatability of analytical methods. The smaller the RSD is, the higher the precision and the better the repeatability is. In this work, the RSD value for precision, repeatability and stability analysis were 0.92%, 0.85%, and 0.73%, which is low enough to indicate the acceptable precision, repeatability and stability. For the sake of strict use of words, we have revised this sentence (Please See, Page 4, Line 137-138).

Reviewer 2 Report

The research article entitled: " Efficient Purification of R-phycoerythrin from Marine Algae (Porphyra yezoensis) based on a Deep Eutectic Solvents Aqueous Two-Phase System  by Y. Xu , Q. Wang , Y. Hou  addresses an interesting issue in the field of "green chemistry"

The study evaluates  the purification of R-phycoerythrin from Marine Algae (Porphyra yezoensis) by DES. The research is of potential interest, but the paper suffers of some limitations.

Major  comments:

The authors have to explain how they purified the protein from DES. In the paper is not reported. Regarding line 194 FT-IR: the use of the word:conformation of R-PE, it is not correct.

The authors reported that the conformation of the R-PE was unchanged.   The Figure showed that the IR DES spectra overlapped the protein bands, suggesting that the "integrity" of protein functional groups are maintained.

Regarding line 99  it is wrong to write ChCL/K2hPO4 (0.8 g/ml, )

line 286: the different amounts of dialyzed R-PE (0.035, 0,040.... mg): how did the authors measure /weigh these quantities?

Minor comments:

TEM images are not clear.

The paper requires a revision of  English language and style

Author Response

Response to Reviewer 2 Comments

Point 1: The research article entitled: " Efficient Purification of R-phycoerythrin from Marine Algae (Porphyra yezoensis) based on a Deep Eutectic Solvents Aqueous Two-Phase System by Y. Xu , Q. Wang , Y. Hou addresses an interesting issue in the field of "green chemistry" The study evaluates the purification of R-phycoerythrin from Marine Algae (Porphyra yezoensis) by DES. The research is of potential interest, but the paper suffers of some limitations.

Response 1: We appreciate very much for the Reviewer’s good comments and kind recommendation. We would try our best to give satisfactory answers in revised version.

We performed a deep analysis of R-PE properties. It was found that the molecular mass and subunit composition (by SDS-PAGE), absorbance peaks (by UV-vis spectra), fluorescence spectral properties (by fluorescence spectra) and secondary structure (by circular dichroism spectra) were accordance with the previous studies. Besides, all spectra results illustrated that the structure of R-PE kept perfect during the whole process, proving the effectiveness of this purification method.

Major comments:

Point 2: The authors have to explain how they purified the protein from DES. In the paper is not reported. Regarding line 194 FT-IR: the use of the word: conformation of R-PE, it is not correct.

Response 2: We appreciate very much for the Reviewer’s useful comments. After extraction, ultrafiltration centrifuge tube (MWCO 10 kDa) was used to filtered the top phase in order to remove DES and K2HPO4. It was revised in the revised version (Please See, Page 10, Line 305-307).

Besides, the “conformation” was replaced by “structure” in the revised version (Please See, Page 7, Line 202).

Point 3: The authors reported that the conformation of the R-PE was unchanged. The Figure showed that the IR DES spectra overlapped the protein bands, suggesting that the "integrity" of protein functional groups are maintained.

Response 3: We appreciate very much for the Reviewer’s useful suggestion. It was amended in the revised version (Please See, Page 7, Line 201-202).

Point 4: Regarding line 99 it is wrong to write ChCL/K2hPO(0.8 g/ml, ).

Response 4: We appreciate very much for the Reviewer’s useful comment. This was revised in the revised version (Please See, Page 3, Line 111-112).

Point 5: line 286: the different amounts of dialyzed R-PE (0.035, 0,040.... mg): how did the authors measure/weigh these quantities?

Response 5: We appreciate very much for the very helpful comments. The amount of the dialyzed R-PE was calculated by the concentration of dialyzed R-PE (calculated by equation (3)) multiplied by R-PE volume.

Minor comments:

Point 6: TEM images are not clear.

Response 6: We appreciate very much for the Reviewer’s good comments. We adjusted the clarity of TEM images in the revised version (Please See, Page 8, Figure 9).

Point 7: The paper requires a revision of English language and style.

Response 7: We appreciate very much for the Reviewer’s useful comment. Before resubmitting, most of the English usages were revised by MDPI English Editing Services (English Editing ID: english-22569). Then, the remaining questions were revised carefully in this revised version.

Reviewer 3 Report

The current manuscript deals with the Efficient Purification of R-phycoerythrin from Marine Algae (Porphyra yezoensis) based on a Deep Eutectic Solvents Aqueous Two-Phase System. It could be od scientific interest and fit within the Journal scope. Only two minor comments:

Acronyms shouls be avoided in the abstract section

Keywords included in the title should be avoided in this section to expand the article visibility

Statistical analysis ANOVA, should be performed and the corresponding significances included in the manuscript.

Author Response

Response to Reviewer 3 Comments

Point 1: The current manuscript deals with the Efficient Purification of R-phycoerythrin from Marine Algae (Porphyra yezoensis) based on a Deep Eutectic Solvents Aqueous Two-Phase System. It could be od scientific interest and fit within the Journal scope. Only two minor comments:

Response 1: We appreciate very much for the Reviewer’s good comments and kind recommendation. We would try our best to give satisfactory answers in revised version.

Point 2: Acronyms shouls be avoided in the abstract section.

Response 2: We appreciate very much the very helpful comments. The abstract section was checked carefully to make sure that every abbreviation has a full term when it appeared for the first time. (Please See, Page 1, Line 21-22, 27).

Point 3: Keywords included in the title should be avoided in this section to expand the article visibility.

Response 3: We appreciate very much the very helpful comments. Keywords were adjusted in the revised version (Please See, Page 1, Line 30).

Point 4: Statistical analysis ANOVA, should be performed and the corresponding significances included in the manuscript.

Response 4: We appreciate very much for the Reviewer’s useful comment. This was revised in the revised version (Please See, Page 4, Figure 3 and Page 11, Line 355-358).

Round 2

Reviewer 1 Report

The manuscript by Xu et al. still contains several errors in scientific writing. The paper should be improved for better sense and flow. No explicit limitations or knowledge gaps that why the authors conducted this study were addressed. Also, I don't see the well-defined novelty of this work. To maintain the standard of Marine Drugs, I think the manuscript is not yet ready for publication in this journal. Some of my comments can be found below, which the authors may consider for further revisions of their writing and expression.

Abstract:

1) Porphyra yezoensis is not the only source of R-PE. What is the key problem/novelty of your study, no indication on this issue in the abstract? Why you conducted this work?

2) Are you proposing a new extraction technique, please clarify what is that?

3) How did PAGE band counts give any sense of your result interpretation? Please explain only the key findings of your work.

4) What is TEM? The abstract should not contain undefined abbreviation.

Introduction:

1) Needs to improve for sense and flow. How many marine algae that are currently the sources of R-PE? Why was Porphyra yezoensis chosen for your study?  

2) The literature review is unclear throughout the Introduction. I don't see any limitations or knowledge gaps that are the initiations of this study. The information in Table 2 should be described in the Introduction. The remaining limitations or knowledge gaps and the novelty of this study should be addressed explicitly.

Figures are not carefully prepared. For example,

Figure 1: No explanation about the embed pictures of phase separation shown in the graph. What is the purpose of showing these pictures?

Figure 2: What is the title of the x-axis?

Figure 3: Inconsistence of term/wording used. 

Deep Eutectic Solvent (g)

protein (mg)

Extraction time (min)

Figure 4: The alpha label is unclear.

Author Response

Response to Reviewer 1 Comments

Point 1: The manuscript by Xu et al. still contains several errors in scientific writing. The paper should be improved for better sense and flow. No explicit limitations or knowledge gaps that why the authors conducted this study were addressed. Also, I don't see the well-defined novelty of this work. To maintain the standard of Marine Drugs, I think the manuscript is not yet ready for publication in this journal. Some of my comments can be found below, which the authors may consider for further revisions of their writing and expression.

Response 1: We appreciate very much for the Reviewer’s good comments and kind recommendation. We would try our best to give satisfactory answers in the revised version. The introduction was modified in revised version (Please See, Page 1, Line 34-38; Page 2, Line 56-58, 74-77 and 78-80). The other comments have also been modified.

Point 2: Porphyra yezoensis is not the only source of R-PE. What is the key problem/novelty of your study, no indication on this issue in the abstract? Why you conducted this work?

Response 2: We appreciate very much for the Reviewer’s good comments. Porphyra yezoensis is one of the major seaweeds abundantly cultivated in China. Therefore, it is easy to obtain and the price is cheap. More importantly, it has a higher protein content (47% of dry weight) than other red algae [Trends Food Sci. Technol. 27 (2012) 57-61; Bioresour. Technol. 131 (2013) 21-27], suggesting its potential use as a source of R-PE. Therefore, we conducted this work to obtain R-PE. (Please See, Page 1, Line 15 and 34-38).

Point 3: Are you proposing a new extraction technique, please clarify what is that?

Response 3: In this work, green solvent deep eutectic solvents (DESs) were used to form deep eutectic solvents aqueous two-phase system (DES-ATPS) with K2HPO4 and successfully applied to purify R-PE from Porphyra yezoensis for the first time. It is a new method for R-PE extraction.

Point 4: How did PAGE band counts give any sense of your result interpretation? Please explain only the key findings of your work.

Response 4: We appreciate very much for the Reviewer’s useful comment. It was refined in the revised version (Please See, Page 5, Line 170-174).

Point 5: What is TEM? The abstract should not contain undefined abbreviation.

Response 5: We appreciate very much for the Reviewer’s useful comment. It was revised in the revised version (Please See, Page 1, Line 27).

Introduction:

Point 6: Needs to improve for sense and flow. How many marine algae that are currently the sources of R-PE? Why was Porphyra yezoensis chosen for your study?

Response 6: We appreciate very much for the Reviewer’s useful comments.

The sense and flow were checked carefully and most of the English usages were revised by MDPI English Editing Services (English Editing ID: english-22569) to make sure no more mistakes.

Currently the sources of R-PE were listed in Table 2. Porphyra yezoensis is one of the major seaweeds abundantly cultivated in China and it is easy to obtain and the price is cheap. More importantly, Porphyra yezoensis has a higher protein content (47% of dry weight) than other red algae, suggesting its potential use as a source of R-PE. Therefore, Porphyra yezoensis was chosen to obtain R-PE for my study. (Please See, Page 1, Line 34-38).

Point 7: The literature review is unclear throughout the Introduction. I don't see any limitations or knowledge gaps that are the initiations of this study. The information in Table 2 should be described in the Introduction. The remaining limitations or knowledge gaps and the novelty of this study should be addressed explicitly.

Response 7: We appreciate very much for the Reviewer’s useful comment. We This part was modified in revised version (Please See, Page 1, Line 34-38; Page 2, Line 56-58, 74-77 and 78-80).

Figures are not carefully prepared. For example,

Point 8: Figure 1: No explanation about the embed pictures of phase separation shown in the graph. What is the purpose of showing these pictures?

Response 8: We appreciate very much for the Reviewer’s useful comment. It was modified in the revised version (Please See, Page 3, Line 97-99 and Line 104-105). The purpose of showing these pictures is to make them more visual to display the DES-K2HPO4 aqueous two-phase system.

Point 9: Figure 2: What is the title of the x-axis?

Response 9: We appreciate very much for the Reviewer’s useful suggestion. It was amended in the revised version (Please See, Page 3, Figure 2).

Point 10: Figure 3: Inconsistence of term/wording used.

Deep Eutectic Solvent (g)

protein (mg)

Extraction time (min)

Response 10: We appreciate very much the very helpful comments. The format of terms was corrected in the revised version (Please See, Page 4, Figure 3).

Point 11: Figure 4: The alpha label is unclear.

Response 11: We appreciate very much the very helpful comments. We adjusted the clarity of alpha label and Figure 4 in the revised version (Please See, Page 5, Figure 4).

Reviewer 2 Report

The authors responded appropriately to the questions posed

Author Response

Response to Reviewer 2 Comments

Point 1: The authors responded appropriately to the questions posed.

Response 1: We appreciate very much for the Reviewer’s useful suggestions and feel grateful for the publication agreement.

This manuscript is a resubmission of an earlier submission. The following is a list of the peer review reports and author responses from that submission.

Round 1

Reviewer 1 Report

The research article entitled: Highly Efficient Purification of R-phycoerythrin from Marine Algae (Porphyra yezoensis) based on Deep Eutectic Solvents Aqueous Two-Phase System  by Xu Y. , Wang Q.  and  Yanhua Hou  addresses the use of deep eutectic solvents as "green chemistry" in order to purify R-phycoerythrin (R-PE), a bioactive protein from marine  algae (Porphyra yezoensis).

Although the topic is interesting,the manuscript is poorly structured and suffers of deep limitations.

The description of the methodology  is unclear and confusing.  The authors seem not to get the focus of the experimental design aim.

The major concern is the English language that  is very poor and doesn't allow the full comprehension of text.

The publication in the present  form of the manuscript  is not recommended

Reviewer 2 Report

The manuscript by Xu et al. evaluates an extraction protocol to gain R-phycoerythrin from a marine alga Porphyra yezoensis. The compound itself is not novel and no elucidation about its bioactivities investigated in this paper. The authors compared their findings with the other studies that used either different extraction procedures or algal sources. With the comparison basis, it is difficult to conclude that the extraction protocol conducted is more or less efficient than the others. Critically, the manuscript contains several errors in English usage, which the authors need a thorough revision with support by a native English speaker.

General comments:

All figures and tables should be understandable when they stand alone. Therefore, the details should be described clearly in the figure legends/table footnotes and not contain any undefined abbreviation.

Lines 12-28: Some articles are missing, and containing other errors in English usage.

Line 35: ... belong to the phylum Rhodophyta ...

From line 35, use Pyezoensis, instead of Porphyra yezoensis

Lines 35-36: In the present tense.

Line 37: In the present tense, and should be 'Also,' not 'Therefore.' 

Line 38: In the present tense.

Lines 40-41, and beyond: Contains many errors in English usage.